# Improved isolation of extracellular vesicles by removal of both free proteins and lipoproteins

**Dmitry Ter-Ovanesyan[1†], Tal Gilboa[1,2†], Bogdan Budnik[1], Adele Nikitina[1], Sara Whiteman[1], Roey Lazarovits[1], Wendy Trieu[1], David Kalish[1], George M Church[1,3], David R Walt[1,2,3]***

[1]Wyss Institute for Biologically Inspired Engineering, Boston, United States; [2]Department of Pathology, Brigham and Women's Hospital, Boston, United States; [3]Harvard Medical School, Boston, United States

**Abstract** Extracellular vesicles (EVs) are released by all cells into biofluids such as plasma. The separation of EVs from highly abundant free proteins and similarly sized lipoproteins remains technically challenging. We developed a digital ELISA assay based on Single Molecule Array (Simoa) technology for ApoB-100, the protein component of several lipoproteins. Combining this ApoB-100 assay with previously developed Simoa assays for albumin and three tetraspanin proteins found on EVs (Ter-Ovanesyan, Norman et al., 2021), we were able to measure the separation of EVs from both lipoproteins and free proteins. We used these five assays to compare EV separation from lipoproteins using size exclusion chromatography with resins containing different pore sizes. We also developed improved methods for EV isolation based on combining several types of chromatography resins in the same column. We present a simple approach to quantitatively measure the main impurities of EV isolation in plasma and apply this approach to develop novel methods for enriching EVs from human plasma. These methods will enable applications where high-purity EVs are required to both understand EV biology and profile EVs for biomarker discovery.

***For correspondence:**
dwalt@bwh.harvard.edu

[†]These authors contributed equally to this work

## Editor's evaluation

This study presents a valuable contribution to how we isolate and analyze EVs. The proposed approaches are supported by solid experimental evidence. This work will be of interest to cell biologists working not only with mammalian EVs but also microbial, parasitic, and plant vesicles.

## Introduction

Extracellular vesicles (EVs) are membrane vesicles released by all cells. EVs contain RNA and protein cargo from their cell of origin and are a promising class of biomarkers in biofluids such as plasma (*Shah et al., 2018*). Since EVs are much less abundant than free proteins and lipoproteins in plasma, enriching them without the co-isolation of these impurities remains highly challenging (*Sódar et al., 2016*; *Smolarz et al., 2019*). This is particularly the case for separating EVs and lipoproteins, as these two classes of particles have overlapping size ranges (*Simonsen, 2017*). Developing EV isolation methods is also particularly challenging due to the inability of most methods, such as the commonly used Nanoparticle Tracking Analysis (NTA) to differentiate between EVs and similarly sized lipoproteins (*Johnsen et al., 2019*). Thus, it is difficult to compare EV isolation methods without suitable techniques to quantify both EV yield and lipoprotein content (*Hartjes et al., 2019*).

**Figure 1.** Validation of ApoB-100 Simoa assay. Simoa ApoB-100 assay was validated using: (**A**) Calibration curve using purified ApoB-100 protein. (**B**) Dilutions of human plasma (from three different individuals) to confirm dilution linearity of endogenous ApoB-100. Error bars represent the standard deviation from two technical replicates.

The online version of this article includes the following source data for figure 1:

**Source data 1.** Validation of Simoa ApoB-100 assay.

We have previously developed a framework for comparing EV isolation methods by measuring three tetraspanin transmembrane proteins present on EVs (CD9, CD63, and CD81) and albumin (***Ter-Ovanesyan et al., 2021***). We used the tetraspanins as a way to compare EV yields across different isolation methods and albumin (the most abundant free protein in plasma) as a way to measure protein contamination. To measure these proteins, we used Single Molecule Array (Simoa) technology, a digital ELISA method that results in high sensitivity and a wide dynamic range (***Rissin et al., 2010***).

In this work, we developed a Simoa assay for ApoB-100, the major protein component of several lipoproteins. ApoB-100 is present on low-density lipoproteins (LDL), intermediate-density lipoproteins (IDL), and very low-density lipoproteins (VLDL) (***German et al., 2006***; ***Sniderman et al., 2019***). By combining the new ApoB-100 assay with our previously developed CD9, CD63, CD81, and albumin assays, we could quantify EVs, lipoproteins, and free proteins for each sample on the same platform. We then used these assays to further improve EV isolation methods from plasma, enabling us to separate EVs from lipoproteins and free proteins at purity levels beyond those of previously described methods. We envision these methods to enable applications requiring high EV purity, such as EV proteomics for biomarker discovery from human biofluids.

## Results

To measure lipoproteins, we developed a Simoa assay against ApoB-100, the protein component of lipoproteins LDL, IDL, and VLDL. Simoa is a digital ELISA method where individual immuno-complexes are trapped on magnetic beads that are loaded into microwells that fit, at most, one bead per well. Counting the 'on wells' thus translates to counting individual protein molecules, providing much higher sensitivity than traditional ELISA (***Rissin et al., 2010***). First, we compared a variety of capture and detector antibodies using calibration curves of the purified ApoB-100 protein standard. We then chose the antibody pair with the highest signal-to-background ratio (***Figure 1A***) and validated this assay with dilution linearity experiments in three individual plasma samples (***Figure 1B***). We also performed spike and recovery experiments using a purified protein standard (***Table 1***). We then combined this Simoa assay for ApoB-100 with our previously developed CD9, CD63, CD81, and albumin assays (***Ter-Ovanesyan et al., 2021***;

**Table 1.** Spike and recovery for ApoB-100 assay. Percent recovery of different concentrations of purified ApoB-100 spike added to plasma and measured using the ApoB-100 Simoa assay.

|  | Spike concentration (ng/ml) | Average recovery (%) |
| --- | --- | --- |
| Spike 1 | 5 | 94.7 |
| Spike 2 | 10 | 87.2 |
| Spike 3 | 50 | 90.8 |
| Spike 4 | 500 | 90.6 |

*Norman et al., 2021*) to simultaneously measure EVs, free proteins, and lipoproteins on the same platform.

We first investigated whether EVs can be separated from ApoB-100-containing lipoproteins using existing techniques. We tested EV separation based on size using size exclusion chromatography (SEC) columns with three different resins (Sepharose CL-2B, CL-4B, and CL-6B). We collected 0.5 ml fractions after performing SEC and used Simoa to measure CD9, CD63, CD81, albumin, and ApoB-100 in each fraction (*Figure 2A*). As in our previous work (*Ter-Ovanesyan et al., 2021*), we calculated relative EV yields between different EV isolation methods (*Figure 2B*). To calculate EV yield, we first determined the ratio of each tetraspanin between conditions and then took the average of the three tetraspanin ratios. To then calculate EV purity, we calculated the ratio of EV yield relative to albumin or ApoB-100 levels. We found that although the ratio of EVs relative to ApoB-100 was higher in the resin with the largest pore size, Sepharose CL-2B, this was at the expense of EV yield relative to the other two resins (*Figure 2C*).

Because separating EVs from lipoproteins based on size alone was not fruitful, we attempted to separate EVs from lipoproteins based on other properties. We first investigated the separation of EVs from lipoproteins and albumin using density gradient ultracentrifugation (DGC). We loaded 1 ml of plasma on an iodixanol gradient and performed ultracentrifugation for 16 hr. We then collected 1 ml fractions and used Simoa to measure tetraspanins, albumin, and ApoB-100 in each fraction. We found that we could readily separate EVs from ApoB-100-containing lipoproteins (*Figure 3*) using DGC, although the EV yield was lower than in SEC (*Figure 3—figure supplement 1A*). As DGC is time-consuming and low throughput, we explored other EV isolation methods that would be more suitable for processing clinical samples.

We next considered whether SEC could be modified to maximize EV yield while removing both free proteins and lipoproteins. First, we wanted to assess the absolute recovery of EVs by SEC using Sepharose CL-6B, the resin with the highest EV yield (*Figure 2—figure supplement 1*). We took advantage of Simoa's wide dynamic range and specificity to measure tetraspanin levels in diluted plasma and compared these levels after EV purification from the same batch of plasma by SEC. We also evaluated how various other parameters affected EV recovery by SEC. We found that performing at least one 10 ml phosphate-buffered saline (PBS) wash in-column, instead of simply washing the resin in bulk before making the column, increased the EV yield significantly (*Figure 2—figure supplement 1A*). One potential reason for this result could be that in-column washes are more effective at removing the ethanol in which the resin is supplied. After performing an in-column wash, we were able to achieve >50% EV yield using SEC with Sepharose CL-6B (*Figure 2—figure supplement 1B*), as measured by comparing tetraspanin levels in SEC fractions 7–10 relative to diluted plasma.

We then decided to take advantage of the property that ApoB-100 is positively charged (*Olsson et al., 1991*), while EVs are generally negatively charged (*Brownlee et al., 2014*) to separate EVs from lipoproteins. It has previously been demonstrated that dual-mode chromatography (DMC) columns with a bottom layer of cation exchange resin below Sepharose CL-4B can be used to isolate EVs (*Van Deun et al., 2020*). Since Sepharose CL-6B yields more EVs than Sepharose CL-4B (*Figure 2C*), we constructed DMC columns with a 2-ml cation exchange resin bottom layer and a top layer of 10 ml Sepharose CL-6B. Inspired by the DMC approach of combining different resins in the same column, we also developed a new type of column, Tri-Mode mixed-mode Chromatography (TMC), where we further added Capto Core 700 to the cation exchange resin in the bottom layer. Capto Core 700 is a multimodal chromatography resin that contains porous beads with an inner core layer functionalized with octylamine groups that bind and trap proteins (*Blom et al., 2014*). Thus, we reasoned that having this resin in the bottom layer would 'catch' free proteins that co-isolate with EVs during SEC (*Figure 4A*). By changing the volumes and ratios of the two resins, the depletion of both albumin and ApoB-100 could be tuned to increase purity, although at the cost of EV yield (*Figure 4—figure supplement 1*). To balance EV yield and purity, we settled on a 2:1 ratio of solid cation exchange resin to multimodal resin in the 2 ml bottom layer.

We compared EV purification from plasma using SEC, DMC, and TMC columns. We first used electron microscopy to image EVs from each column. We found that, although lipoproteins were still present, DMC and TMC led to a higher purity of EVs relative to lipoproteins (*Figure 4B*). We then used Simoa to quantify the relative levels of EVs, lipoproteins, and free proteins using SEC, DMC, and TMC columns. We collected fractions 9–12 (instead of fractions 7–10 as for SEC) for DMC and TMC to

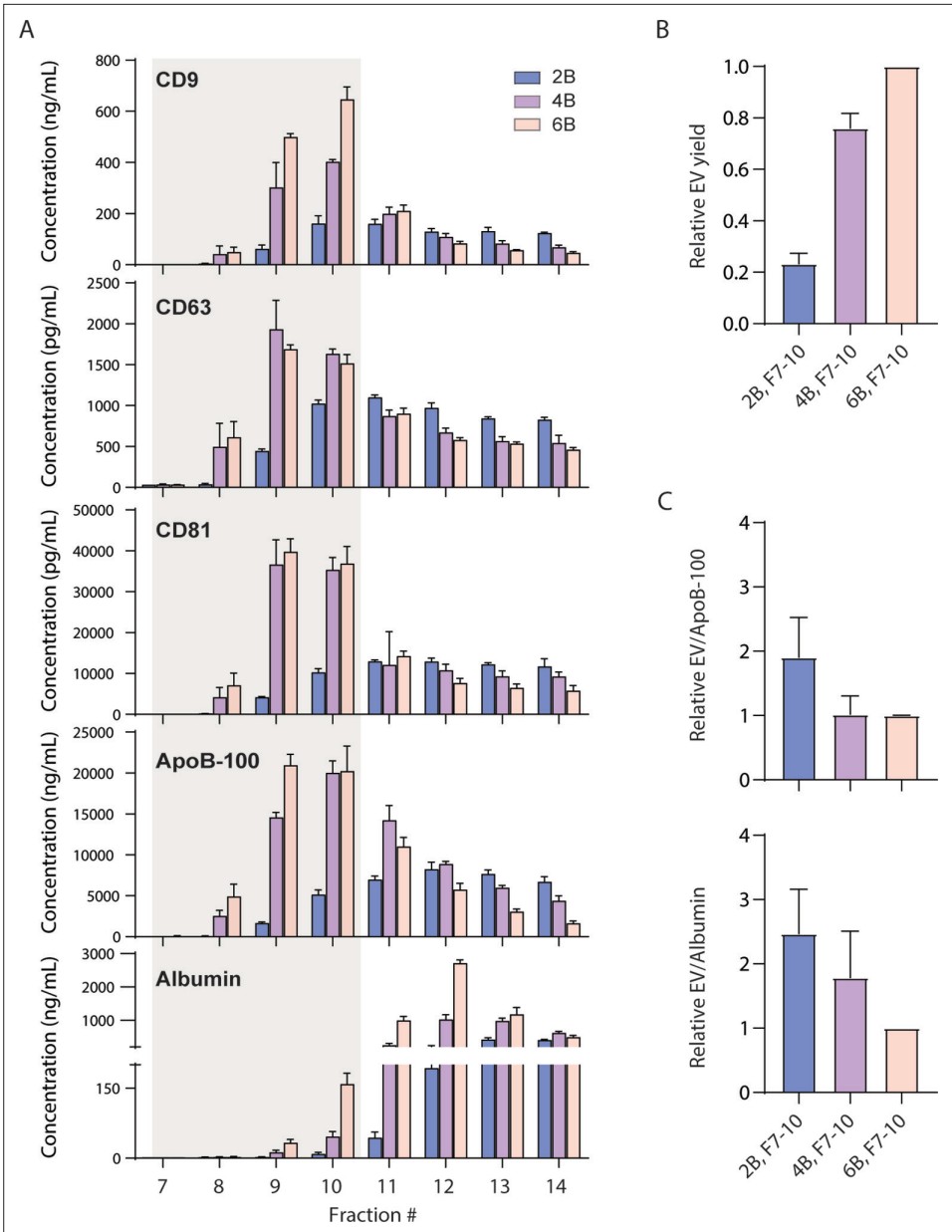

**Figure 2.** Size exclusion chromatography (SEC) of plasma using different resins. (**A**) Levels of CD9, CD63, CD81, ApoB-100, and albumin were measured by Simoa after SEC of 1 ml plasma in each fraction using either Sepharose CL-2B, Sepharose CL-4B, or Sepharose CL-6B resin. (**B**) Extracellular vesicle (EV) yield is calculated in fractions 7–10 for Sepharose CL-2B, Sepharose CL-4B, or Sepharose CL-6B by averaging the ratios of CD9, CD63, and CD81. (**C**) Purity of EVs with respect to lipoproteins or free proteins is calculated by dividing relative EV yield (the average of the ratios of CD9, CD63, and CD81) by levels of ApoB-100 (top) or albumin (bottom). Error bars represent the standard deviation of four columns measured on different days with two technical replicates each.

The online version of this article includes the following source data and figure supplement(s) for figure 2:

**Source data 1.** Simoa data (protein concentrations) for fractions of SEC columns with different resins.

**Source data 2.** Simoa data (protein concentrations) for SEC column with different number of washes.

**Figure supplement 1.** In-column phosphate-buffered saline (PBS) washes improve extracellular vesicle (EV) recovery.

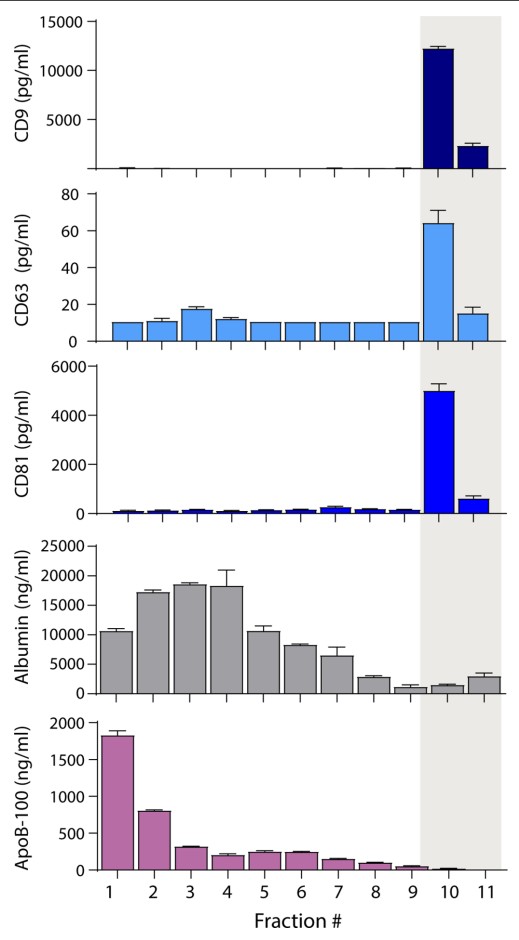

**Figure 3.** Separation of extracellular vesicles (EVs), lipoproteins, and free proteins from plasma using density gradient centrifugation. Levels of CD9, CD63, CD81, albumin, and ApoB-100 were measured by Simoa in individual 1 ml fractions (collected from the top) after density gradient centrifugation of plasma using an iodixanol gradient. Error bars represent the standard deviation of two replicates of each measurement.

The online version of this article includes the following source data and figure supplement(s) for figure 3:

**Source data 1.** Simoa data (protein concentrations) for different density gradient centrifugation fractions.

**Figure supplement 1.** Comparison of density gradient centrifugation to size exclusion chromatography (SEC).

account for the extra 2 ml of resin in the column (*Figure 4—figure supplement 2*). We found that DMC and TMC columns significantly depleted ApoB-100, but also led to some loss in EV yield, particularly CD9, compared to SEC columns (*Figure 4—figure supplement 3*). Calculating the relative yields of each tetraspanin and averaging the three tetraspanin ratios to calculate EV yield, we found that DMC and TMC columns had a lower EV yield than SEC but significantly higher EV/ApoB-100 ratios. Although DMC columns had higher ratios of EVs to ApoB-100 compared to SEC, the ratio of EVs to albumin remained the same. The TMC column, on the other hand, had a higher ratio of both EVs to ApoB-100 and EVs to albumin compared to the SEC column (*Figure 4C–E*).

To assess the utility of highly pure EVs isolated with TMC, we performed mass spectrometry-based proteomic analysis. Performing mass spectrometry on EVs from plasma is challenging because levels of both free proteins and lipoproteins are several orders of magnitude higher than those of EV proteins (*Smolarz et al., 2019*). Using TMC, we were able to detect 780 proteins from EVs isolated from only 1 ml of plasma (*Supplementary file 1*). These results demonstrate the advantage of using TMC for deep proteomics analysis using a small sample volume.

Single-use chromatography columns, whether SEC to maximize EV yield or TMC to maximize EV purity, represent an attractive platform for isolating EVs from clinical samples as they are inexpensive and take a short time to run (*Monguió-Tortajada et al., 2019*). The throughput of columns, however, is limited. Columns are usually run one at a time, and although it is possible to run more than one column at the same time, this becomes challenging if done manually. To increase the throughput and reproducibility of column-based EV isolation, we built a semi-automated stand for running eight columns in parallel. Using a syringe pump run by a Raspberry Pi, we were able to dispense liquid to all columns in parallel (*Figure 5A, B*). We tested the reproducibility of our device using Simoa and found high concordance between eight SEC columns run by the device and eight SEC columns run manually for EV isolation from plasma (*Figure 5C*). Although this device was built as a proof of principle to run eight columns in parallel, we envision building a similar device that could run many more columns simultaneously in the future.

## Discussion

In this work, we developed methods to enrich EVs from both lipoproteins and free proteins in plasma based on our ability to measure proteins associated with these different components using ultrasensitive assays. First, we developed and validated a Simoa assay for ApoB-100. We then combined this

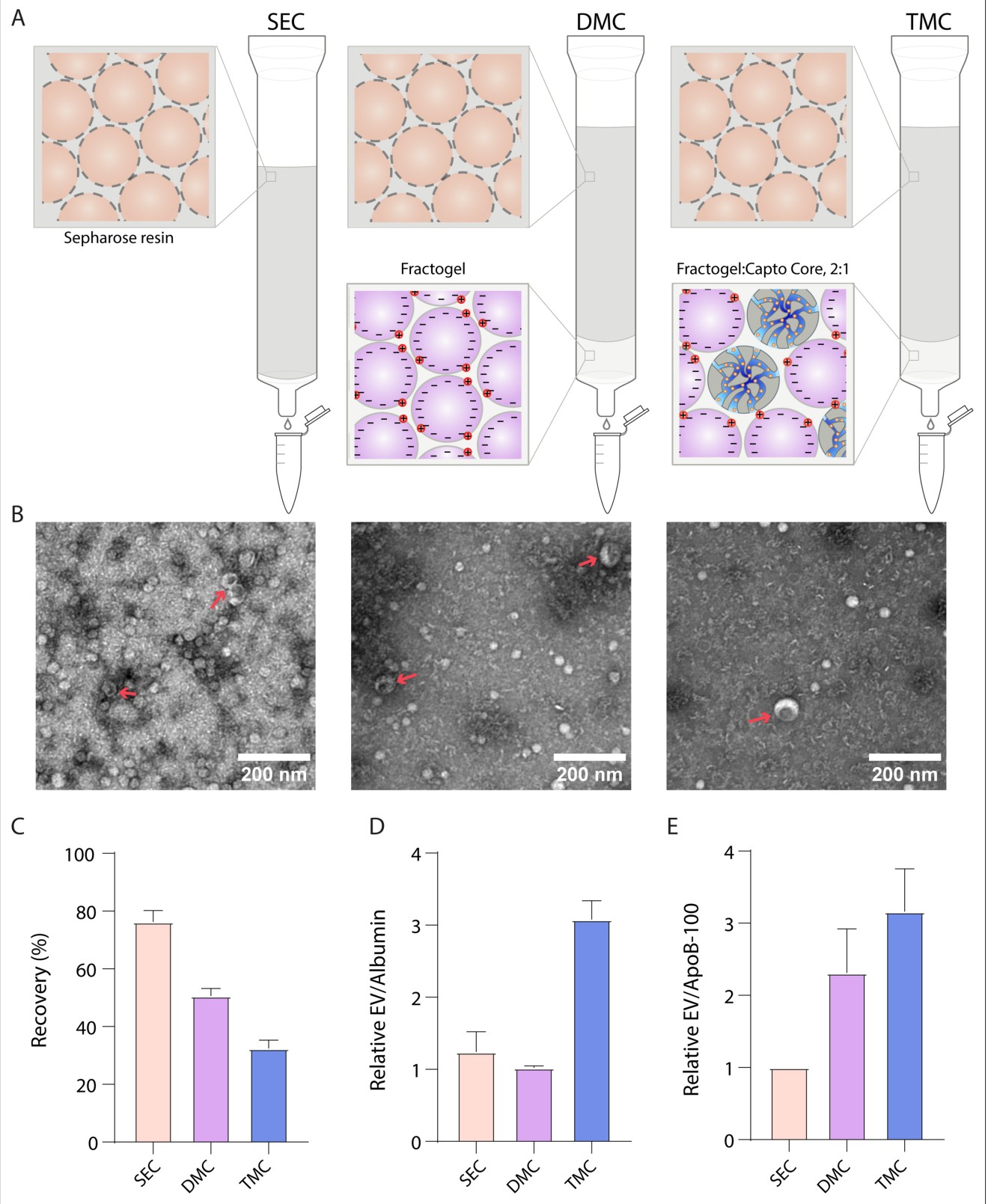

**Figure 4.** Comparison of novel columns for extracellular vesicle (EV) isolation from plasma using electron microscopy and Simoa. (**A**) Schematic of the columns being compared: size exclusion chromatography (SEC) column comprised of 10 ml Sepharose CL-6B, dual-mode chromatography (DMC) columns comprised of 10 ml Sepharose CL-6B SEC resin atop 2 ml Fractogel cation exchange resin, Tri-Mode Chromatography (TMC) columns comprised of 10 ml Sepharose CL-6B SEC resin atop 2 ml 2:1 ratio of 2 ml Fractogel cation exchange resin to Capto Core 700 multimodal

*Figure 4 continued on next page*

*Figure 4 continued*

chromatography resin. (**B**) Electron microscopy of EVs isolated from plasma using SEC (left), DMC (middle), or TMC (right) columns. EVs indicated with red arrows (among background of lipoproteins). (**C**) EV recovery is calculated for EV isolation from plasma for SEC (fractions 7–10), DMC (fractions 9–12), or TMC (fractions 9–12). Simoa measurements in the designated fractions for CD9, CD63, and CD81 are taken as a ratio relative to measurements of these proteins from diluted plasma and these three ratios are then averaged to calculate recovery. (**D**) Purity of EVs with respect to free proteins is determined by dividing relative EV yield (the average of the ratios of CD9, CD63, and CD81) by relative levels of albumin in each condition. (**E**) Purity of EVs with respect to lipoproteins is determined by dividing relative EV yield (the average of the ratios of CD9, CD63, and CD81) by relative levels of ApoB-100 in each condition. Error bars represent the standard deviation of four column measured on different days with two technical replicates each.

The online version of this article includes the following source data and figure supplement(s) for figure 4:

**Source data 1.** Simoa data (protein concentrations) for TMC columns with different ratios of resins in the bottom layer.

**Source data 2.** Simoa data (protein concentrations) for different fractions of SEC and DMC columns.

**Source data 3.** Simoa data (protein concentrations) comparing SEC (fractions 7-10), DMC and TMC columns (fractions 9-12).

**Figure supplement 1.** Comparison of resin volumes and ratios for Tri-Mode Chromatography (TMC) column.

**Figure supplement 2.** Analysis of markers in individual fractions of size exclusion chromatography (SEC) and dual-mode chromatography (DMC).

**Figure supplement 3.** Comparison of marker levels in size exclusion chromatography (SEC), dual-mode chromatography (DMC), and Tri-Mode Chromatography (TMC).

assay with previously developed Simoa assays for the tetraspanins CD9, CD63, and CD81, as well as albumin (*Ter-Ovanesyan et al., 2021*; *Norman et al., 2021*). With these assays in place, we were able to quantify EVs, free proteins, and lipoproteins from the same sample on one experimental platform. Using this approach, we assessed different ways of separating EVs from lipoproteins with the aim of developing improved EV isolation methods.

Plasma contains several types of lipoproteins with varying protein and lipid compositions. Although there is not a single present on all lipoproteins, we chose to measure ApoB-100, as it is a protein component of several lipoproteins (such as LDL, IDL, and VLDL) that overlap in size with EVs (*Simonsen, 2017*; *Johnsen et al., 2019*). We evaluated the possibility that SEC using resins with three different pore sizes might separate EVs from ApoB-100-containing lipoproteins using our platform. We previously used our tetraspanin and albumin Simoa assays to directly compare EV yield and free protein contamination for different SEC resins (*Ter-Ovanesyan et al., 2021*). Here, we used a similar approach to compare EV yield and lipoprotein contamination by including the ApoB-100 assay and found that we were unable to effectively separate tetraspanins from ApoB-100 by SEC. We also used our Simoa assays to evaluate DGC and showed that this technique enables good separation of tetraspanins from ApoB-100 and albumin; however, since DGC requires an ultracentrifuge, is low throughput, and time-intensive, it is not suitable for clinical samples.

We used our assays to develop novel methods for separating EVs from lipoproteins. A previous study described DMC columns that deplete lipoproteins by combining SEC using Sepharose CL-4B with a second bottom layer of cation exchange resin (*Van Deun et al., 2020*). We modified DMC to include the higher yield Sepharose CL-6B resin and demonstrated depletion of most of the ApoB-100, although at the cost of some EV depletion. To improve the ratio of EVs to ApoB-100 and albumin, we developed a new method that combines a top layer of Sepharose CL-6B with a bottom layer of both cation exchange resin and a multimodal chromatography resin called Capto Core 700. These 'Tri-Mode mixed-mode Chromatography', or TMC columns, produced EV preparations of higher purity relative to SEC columns in terms of both their lower albumin and lipoprotein content. As the multimodal chromatography resin binds free proteins but not EVs, the TMC columns reduce the co-elution of free protein with EVs.

This work presents a framework for quantitatively comparing EV isolation methods. There is not a single optimal way to isolate EVs because the purification method must be matched to the application; therefore, it is crucial to have effective ways of comparing both the yield and purity of different isolation methods. We developed TMC columns for applications where EVs of very high purity are needed and optimized these columns for EV purification from plasma using our Simoa assays. We envision these columns will be particularly useful for EV biomarker discovery using proteomics, where

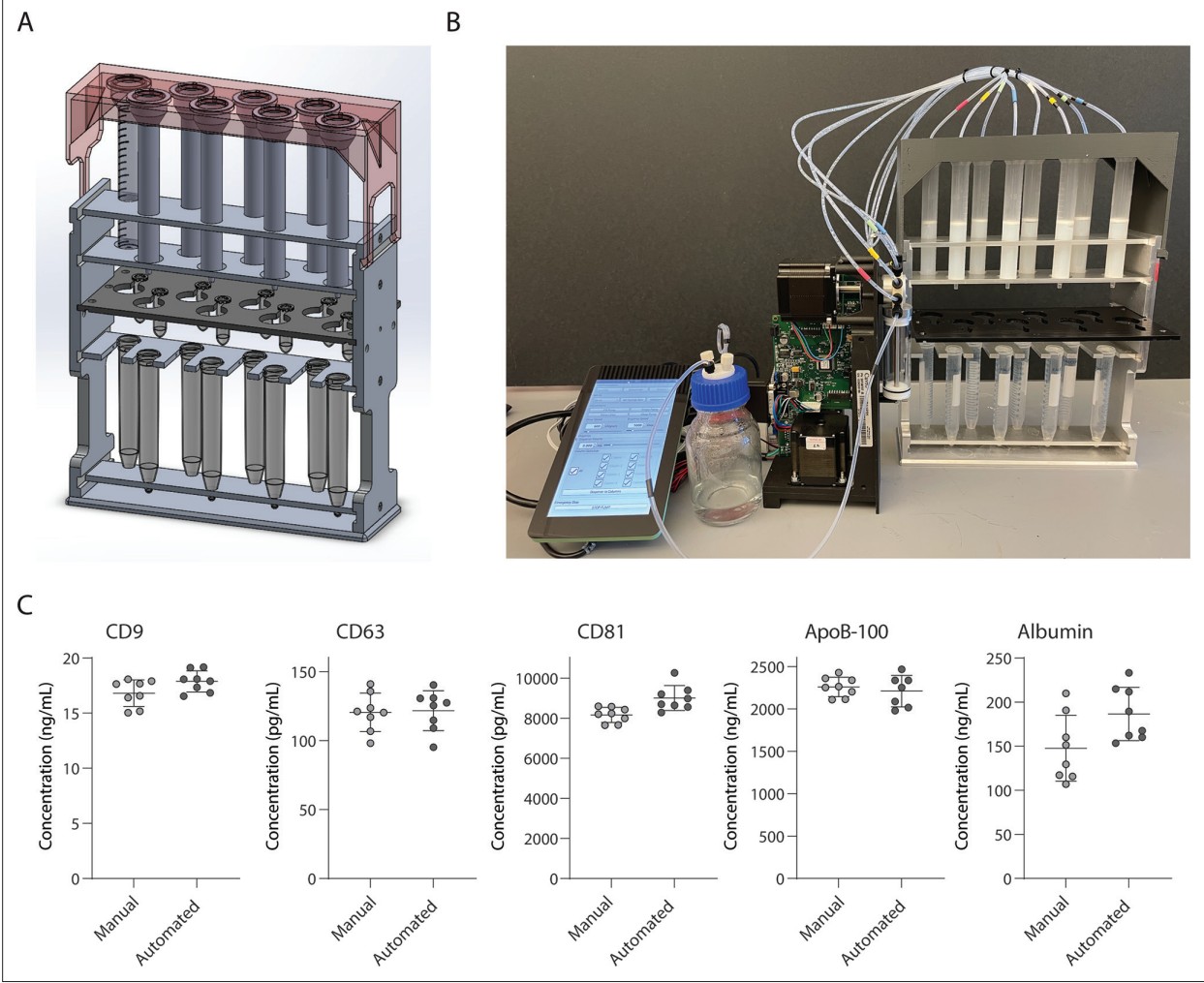

**Figure 5.** Development and validation of automated device for running size exclusion chromatography (SEC) columns in parallel. (**A**) CAD image of semi-automated SEC stand designed to hold eight columns at once with sliding collection tube holder that allows liquid to drip either into 2 ml collection tubes, or to waste. (**B**) Photograph of stand connected to a Tecan Cavro syringe pump controlled by a Raspberry Pi. (**C**) Simoa comparison of CD9, CD63, CD81, ApoB-100, and albumin when SEC was performed on 16 samples of 1 ml plasma using either manual SEC (8 samples) or SEC on the automated device (8 samples). Each point is the average of two Simoa measurements (technical replicates).

The online version of this article includes the following source data for figure 5:

**Source data 1.** Simoa data (protein concentrations) comparing manual and automated SEC EV isolation (fractions 7-10).

EV contamination with lipoproteins and free proteins prevents deep coverage (*Smolarz et al., 2019*). A recent study using a three-step protocol (polyethylene glycol precipitation followed by iohexol gradient fractionation and SEC) to enrich EVs from 1 ml plasma reported the detection of 250 proteins (*Zhang et al., 2020*). Another study reported the detection of 1187 proteins from plasma using ultracentrifugation, density gradient, and then SEC; the starting volume in that study was 40–80 ml plasma (*Karimi et al., 2018*). Using TMC columns, we were able to measure the plasma EV proteome using an easy, single-step isolation protocol and detect 780 proteins using a 1-ml sample. By also building an automated device for running columns in parallel, we demonstrate a path toward using column-based methods for clinical samples. Future iterations of the device will further increase the

sample throughput. Taken together, the methods we developed will contribute to the realization of EV profiling in molecular diagnostics.

# Methods

## Key resources table

| Reagent type (species) or resource | Designation | Source or reference | Identifiers | Additional information |
|---|---|---|---|---|
| Biological sample (human) | Plasma | BioIVT | Cat# HUMANPLK2PNN | Pooled gender, K2EDTA |
| Antibody | anti-CD9 (rabbit monoclonal) | Abcam | Cat# ab263024 | Simoa capture (0.031 µg per assay) |
| Antibody | anti-CD9 (mouse monoclonal) | Abcam | Cat# ab58989 | Simoa detector (0.06 µg per assay) |
| Antibody | anti-CD63 (mouse monoclonal) | R&D Systems | Cat# MAB5048 | Simoa capture (0.031 µg per assay) |
| Antibody | anti-CD63 (mouse monoclonal) | BD | Cat# 556019 RRID: AB_396297 | Simoa detector (0.0435 µg per assay) |
| Antibody | anti-CD81 (mouse monoclonal) | Abcam | Cat# ab79559 | Simoa capture (0.031 µg per assay) |
| Antibody | anti-CD81 (mouse monoclonal) | BioLegend | Cat# 349502 RRID: AB_10643417 | Simoa detector (0.0435 µg per assay) |
| Antibody | Human Serum Albumin DuoSet ELISA | R&D Systems | Cat# DY1455 | Simoa capture (0.031 µg per assay) and detector (0.002 µg per assay) |
| Antibody | anti-ApoB (mouse monoclonal) | R&D Systems | Cat# mab4124 RRID:AB_2057095 | Simoa capture (0.031 µg per assay) |
| Antibody | anti-ApoB (mouse monoclonal) | R&D Systems | Cat# mab41242 | Simoa detector (0.08 µg per assay) |
| Peptide, recombinant protein | CD9 | Abcam | Cat# ab152262 | |
| Peptide, recombinant protein | CD63 | Origene | Cat# TP301733 | |
| Peptide, recombinant protein | CD81 | Origene | Cat# TP317508 | |
| Peptide, recombinant protein | Albumin | Abcam | Cat# ab201876 | |
| Other | Purified ApoB-100 Standard | Origene | Cat# BA1030 | Protein standard for Simoa |
| Other | Sepharose CL-2B | Cytiva | Cat# 17014001 | Resin for SEC |
| Other | Sepharose CL-4B | Cytiva | Cat# 17015001 | Resin for SEC |
| Other | Sepharose CL-6B | Cytiva | Cat# 17016001 | Resin for SEC |
| Other | Fractogel EMD SO3- (M) | MilliporeSigma | Cat# 1168820100 | Resin for DMC/TMC |
| Other | Capto Core 700 multimodal chromatography resin | Cytiva | Cat# 17548102 | Resin for TMC |
| Other | Econo-Pac Chromatography Columns | Bio-Rad | Cat # 7321010 | Empty columns |

## Human samples

Pooled human plasma (collected in K2 EDTA tubes) was ordered from BioIVT. Plasma was thawed at room temperature and centrifuged at 2000 × *g* for 10 min. The supernatant was filtered through a 0.45-µm Corning Costar Spin-X centrifuge tube (MilliporeSigma) at 2000 × *g* for 10 min. For all direct comparison experiments, plasma was first pooled and 1 ml used per EV isolation.

## Simoa assays

Simoa assays for CD63, CD81, and albumin were performed as previously described (*Ter-Ovanesyan et al., 2021*; *Norman et al., 2021*). Due to antibody availability, CD9 ab263024 (Abcam) was used as a capture antibody instead of ab195422 (Abcam). For ApoB-100, mab4124 (R&D Systems) was used as the capture antibody, mab41242 (R&D Systems) was used as the detector antibody, and purified ApoB-100 BA1030 (Origene) was used as a standard. For SEC, onboard dilution was performed with 4× dilution for each of the assays, with an additional 4× off-board dilution for CD9 and 10× off-board dilution for ApoB-100. For measuring protein levels in total plasma, each protein was measured with 4× onboard dilution and three additional off-board dilutions: for CD9 – 40×, 80×, and 160×; for CD63 and CD81 – 3×, 9×, and 27×; for albumin – 100×, 3000×, and 9000×; and for ApoB-100 – 100×, 300×, and 900× dilution. All samples were measured in duplicate using the HD-X analyzer (Quanterix). Tetraspanins were measured with a two-step assay, while albumin and ApoB-100 were measured with a three-step assay. Average Enzyme per Bead (AEB) values were calculated by the HD-X software.

## Calculation of EV yield and purity

EV yield was calculated for EV isolation from plasma for SEC (fractions 7–10), DMC (fractions 9–12), or TMC (fractions 9–12). Levels of the three tetraspanins CD9, CD63, and CD81 in the designated EV fractions and their levels in total plasma were measured using Simoa. The yield of each tetraspanin was calculated by dividing its level in the EV fraction by its level in total plasma. The total EV yield was then calculated as the average of the three ratios. Relative EV yield for comparing multiple conditions was calculated by dividing the EV yield of each condition by the highest EV yield of all the conditions. The purity of EVs with respect to free proteins or lipoproteins was determined by dividing the relative EV yield by relative levels of albumin or ApoB-100 in the EV fractions.

## Validation of ApoB-100 Simoa assay

Antibodies were first cross-tested using serial dilutions of purified protein standard. The antibody pair with the highest signal-to-background ratio was chosen. The assay was validated using dilution linearity and spike and recovery experiments. Plasma samples were diluted serially in the assay-specific buffer, a dilution factor in the middle of the linear range was chosen to be the dilution factor for the spike and recovery test. Three protein concentrations of purified ApoB-100 were spiked into the diluted plasma from the top calibrator used in the calibration curve. All recoveries fell in the range of 85–100% (*Table 1*). The assay validation was conducted using commercially available plasma samples (BioIVT).

## Preparation of columns

Sepharose CL-2B, Sepharose CL-4B, and Sepharose CL-6B resins (Cytiva) were washed with PBS in a glass bottle. The volume of resin was washed three times with an equal volume of PBS before use. Econo-Pac Chromatography columns (Bio-Rad) were packed with resin and a frit was inserted into the column above the resin. For all columns in *Figures 4 and 5*, each column was washed with 10 ml PBS (twice 5 ml at a time) prior to loading of sample. For SEC columns, resin was added until the bed volume (resin without liquid) reached 10 ml. For DMC columns, Fractogel EMD SO3- (M) (MilliporeSigma) was added as a bottom layer with 2 ml bed volume, and 10 ml of Sepharose CL-6B bed volume was added as a top layer. For TMC columns, a 2:1 by volume (of dry resin) mixture was prepared of Fractogel EMD SO3- (M) (MilliporeSigma) and Capto Core 700 (Cytiva) and 2 ml bed volume bottom layer was added to the column before 10 ml of Sepharose CL-6B bed volume was added as a top layer.

## Collection of column fractions

Sample (1 ml plasma) was loaded once PBS from wash had finished going through the column. Once the sample fully entered the column, 0.5 ml fractions were collected. PBS was then added in volumes equal to those being collected for one fraction (0.5 ml) or a set of four fractions (2 ml). In experiments where just the EV fractions were collected, fractions 7–10 were collected for SEC and fractions 9–12 were collected for DMC and TMC.

## Density gradient centrifugation

DGC was performed as previously described (*Norman et al., 2021*). Four layers of OptiPrep (iodixanol) were prepared and stacked in a 13.2-ml polypropylene tubes (Beckman Coulter) from bottom to

top: 3 ml 40%, 3 ml 20%, 3 ml 10%, and 2 ml 5%. OptiPrep (MilliporeSigma) was diluted in a solution of 0.25 M sucrose (MilliporeSigma) and pH 7.4 Tris–EDTA (MilliporeSigma). Sample (1 ml plasma) was loaded on top of the gradient and centrifuged at 100,000 RCF in an SW 41 Ti swinging bucket rotor for 18 hr at 4°C using a Beckman Coulter Optima XPN-80 ultracentrifuge. After centrifugation, fractions were removed from the top 1 ml at a time. For the DGC, SEC, and DGC–SEC comparisons, fraction 10 was analyzed (directly for the DGC condition, or then run through an SEC column for DGC–SEC condition).

## Negative staining and transmission electron microscopy (TEM) imaging

Carbon-coated grids (CF-400CU, Electron Microscopy Sciences) were glow discharged, and 5 µl of the sample was absorbed to the grid for 1 min. Excess sample was blotted with a Whatman paper. The grid was then stained with 5 µl 1% uranyl acetate for 15 s and excess stain was blotted. Samples were imaged on a JEOL 1200EX – 80 kV transmission electron microscope with an AMT 2k CCD camera.

## Mass spectrometry

EVs were isolated from 1 ml plasma using TMC columns with a 2-ml bed volume bottom layer of 2:1 of Fractogel EMD SO3- (M) (MilliporeSigma) to Capto Core 700 (Cytiva) and 10-ml bed volume top layer of Sepharose CL-6B (Cytiva). EVs were concentrated using Amicon Ultra-2 centrifugal 10 kD filter (MilliporeSigma). After concentration, EV protein was precipitated by adding 9 volumes of 100% ethanol to 1 volume of EVs, vortexing and leaving at −20°C for 30 min. Sample was then centrifuged at $16,000 \times g$ for 15 min at 4°C. Supernatant was removed and pellet was left to air dry for 10 min. Sample was then sent to Bruker for proteomics analysis. Sample was resuspended in 50 mM triethylammonium bicarbonate (Thermo Fisher Scientific) and digested for 2 hr at 50°C using Trypsin Platinum (Promega) using 1:50 Trypsin to sample ratio by mass. After evaporating solution in a Vacufuge (Eppendorf) ar 45°C, sample was resolubilized in 10 µl 0.1% formic acid (Thermo Fisher Scientific). Next, 1.5 µl of sample was injected into C18 tips (Evosep) and eluted into a 25-cm length 150 µm internal diameter PepSep analytical column packed with 1.5 µm C18 beads (Dr. Maisch). Sample was eluted into a Bruker timsTOF HT. A gradient from 3% of to 28% of 0.1% formic acid in acetonitrile at 63 min was then increased to 85% until 80 min. Data were analyzed using Spectronaut 17 (Biognosys) software for data-independent acquisition (DIA). A false discovery rate of 1% was used at both the peptide and protein levels. Keratin proteins (likely contaminants) were manually removed from the list.

## SEC stand and automated device

Experiments in *Figure 5* were performed using the automated SEC stand connected to a Cavro XLP 6000 syringe pump (Tecan) controlled by a Raspberry Pi 4 model B. CAD files and instructions for assembly are provided in the Supplementary Information. Code for Raspberry Pi is deposited at: https://github.com/Wyss/automated-chromatography/, (*Kalish and Tat, 2023*). All other SEC experiments were performed using custom SEC stand described previously (*Ter-Ovanesyan et al., 2021*).

# Acknowledgements

The authors thank John Wilson for suggestions regarding the TMC column, Allen Tat for help with programming the software for the Raspberry Pi, Jan Van Deun for helpful discussions, and Matt Willets, Diego Assis, and Elizabeth Gordon at Bruker for help with mass spectrometry. This work was funded by Chan Zuckerberg Initiative, Good Ventures, and the Wyss Institute. Tal Gilboa is an awardee of the Weizmann Institute of Science Women's Postdoctoral Career Development Award.

# Additional information

### Competing interests

Dmitry Ter-Ovanesyan: DT has filed IP on methods for EV isolation and analysis. Tal Gilboa: TG has filed IP on methods for EV isolation and analysis. Bogdan Budnik: BB has filed IP on methods for EV isolation and analysis. David Kalish: DK has filed IP on methods for EV isolation and analysis. George

M Church: GMC has filed IP on methods for EV isolation and analysis. David R Walt: DRW has filed IP on methods for EV isolation and analysis. The other authors declare that no competing interests exist.

## Funding

| Funder | Grant reference number | Author |
|---|---|---|
| Chan Zuckerberg Initiative | NeuroDegeneration Challenge Network Grant 2018-191864 | Dmitry Ter-Ovanesyan |
| Good Ventures Foundation | | Dmitry Ter-Ovanesyan |
| Wyss Institute | | Dmitry Ter-Ovanesyan |
| Weizmann Institute of Science | Women's Postdoctoral Career Development Award | Tal Gilboa |

The funders had no role in study design, data collection, and interpretation, or the decision to submit the work for publication.

## Author contributions

Dmitry Ter-Ovanesyan, Conceptualization, Investigation, Methodology, Writing – original draft, Writing – review and editing; Tal Gilboa, Investigation, Methodology, Writing – original draft, Writing – review and editing; Bogdan Budnik, Adele Nikitina, Sara Whiteman, Roey Lazarovits, Wendy Trieu, Investigation, Methodology, Writing – review and editing; David Kalish, Resources, Software, Methodology, Writing – review and editing; George M Church, Supervision, Funding acquisition, Project administration, Writing – review and editing; David R Walt, Supervision, Funding acquisition, Writing – original draft, Project administration, Writing – review and editing

## Author ORCIDs

Dmitry Ter-Ovanesyan 
Tal Gilboa 
Roey Lazarovits 
David R Walt 

## Decision letter and Author response

Decision letter https://doi.org/10.7554/eLife.86394.sa1
Author response https://doi.org/10.7554/eLife.86394.sa2

# Additional files

## Supplementary files

• Supplementary file 1. Protein list identified by mass spectrometry of extracellular vesicles (EVs) isolated from plasma using Tri-Mode Chromatography (TMC).

• MDAR checklist

## Data availability

All data generated or analyzed during this study are included in the manuscript and supporting files; mass spectrometry data Source Data files have been provided.

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
