## [Editor Report]

This study presents a valuable contribution to how we isolate and analyze EVs. The proposed approaches are supported by solid experimental evidence. This work will be of interest to cell biologists working not only with mammalian EVs but also microbial, parasitic, and plant vesicles.

---

## [Decision Letter]

**Decision letter after peer review:**

Thank you for submitting your article "Improved Isolation of Extracellular Vesicles by Removal of Both Free Proteins and Lipoproteins" for consideration by *eLife*. Your article has been reviewed by 3 peer reviewers, and the evaluation has been overseen by a Reviewing Editor and Suzanne Pfeffer as the Senior Editor. The following individuals involved in the review of your submission have agreed to reveal their identity: James Byrd (Reviewer #2); Qing Zhou (Reviewer #3).

Essential revisions:

1) Please observe that, due to the methodological nature of your study, it is extremely important to make sure that you consider the experimental suggestions provided by reviewers 1-3;

2) We expect this manuscript to be of use to a wide audience of biologists since the EV community is growing rapidly. Please also consider the suggestions for improving the manuscript's presentation and discussion depth.

*Reviewer #1 (Recommendations for the authors):*

The paper can be very useful, but can be improved by addressing a few comments:

1. 'We then decided to take advantage of the property that ApoB-100 is positively charged (12), while EVs are generally negatively charged (13) to separate EVs from lipoproteins.'

LDL particles, as particles, are negatively charged. See, for example:

• La Belle M, Blanche PJ, Krauss RM. Charge properties of low-density lipoprotein subclasses. J Lipid Res. 1997 Apr;38(4):690-700. PMID: 9144084.

• Reynolds L, Mulik RS, Wen X, Dilip A, Corbin IR. Low-density lipoprotein-mediated delivery of docosahexaenoic acid selectively kills murine liver cancer cells. Nanomedicine (Lond). 2014 Jul;9(14):2123-41. doi: 10.2217/nnm.13.187. Epub 2014 Jan 7. PMID: 24397600; PMCID: PMC4156561.

The paper cited shows the charge of certain ApoB100 peptides!

LDL, VLDL, etc., contain many other components, not just ApoB100.

2. 'We compared EV isolation from plasma using SEC, DMC, and TMC columns. We first used electron microscopy to image EVs from each column and found that TMC led to EVs of the highest purity (Figure 4B).'

The majority of the particles present on TEM images are lipoproteins, more likely VLDL or Chylomicrons. One can notice more dense round shaped particles, different from 'doughnut-shaped' EVs. For example, Figure 4B TMC column TEM image has a single EV, while eleven lipoprotein particles are visible.

3. 'Using TMC, we were able to detect 780 proteins from EVs isolated from only 1 ml of plasma (Supplementary table 1). These results demonstrate the advantage of using TMC for deep proteomics analysis using a small sample volume.'

According to the dataset provided in the supplementary materials, ApoB100 remains the most abundant protein, as well as other lipoproteins, including the ones specific for High-density lipoproteins. Also, a lot of free proteins were identified. In my experience, 780 proteins after albumin depletion are not as high for raw plasma proteomics analysis.

4. 'In this work, we developed methods to separate EVs from both lipoproteins and free proteins in plasma based on our ability to measure proteins associated with these different components using ultrasensitive assays.'

This statement is not confirmed by proteomics analysis and TEM images. Authors can use the more accurate term 'enriched EVs' compared to lipoproteins. Also, depletion of chylomicrons and HDL were not shown.

*Reviewer #2 (Recommendations for the authors):*

I have a few suggestions for the authors' consideration:

It may be more accurate to describe the free proteins and lipoproteins as impurities (i.e., present at the outset, but not intended to be present in the final preparation), rather than contaminants (i.e., introduced inadvertently)

The introduction could be more laser-focused on plasma since that's the focus of the work, and the lipoprotein issue may be greatly magnified compared to in e..g, urine

A few more words about the principle of the Simoa assay could be useful to the reader

"These results demonstrate the advantage of using TMC for deep proteomics analysis using a small sample volume." Some comparison to the prior state of the art would bolster this conclusion.

Beyond the relative yields, can synthetic EV spike-ins be used to calculate the absolute yields?

Figure legend for Figure 2: The EV yield is calculated "by averaging the ratios of CD9, CD63, and CD81." I think this be explained in clearer terms. This is the average of what is taken in ratio to what? This comment applies to a variety of other places in the paper where ratios are mentioned but without precise definitions of the numerator and denominator.

"Simoa measurements in the designated fractions for CD9, CD63, and CD81 are taken as a ratio relative to measurements of these proteins from diluted plasma and these three rations are then averaged to calculate recovery." Here again, I think this can be explained with a clearer description of what is in the numerator and denominator of these ratios (also a typo, "rations")

It would be helpful if the authors wrote down the calculation they're referring to here: "by averaging the ratios of each of the tetraspanin levels between conditions" The numerator and denominator in the ratios are not as clear as they ideally could be.

A challenge is that after the removal of the main known impurities, whether additional proteins identified in proteomics experiments are in the vesicles or free proteins is not simple to distinguish. How does the relative quantitation of proteins after the application of the new column compare with plasma proteomics performed without enrichment for extracellular vesicles? Is the rank order of abundance similar?

The authors have shared all the relevant data, as well as additional resources that will be useful to the scientific community.

*Reviewer #3 (Recommendations for the authors):*

1) The "relative EV yields" was used to quantify the amount of EVs across different methods. The authors described it as "the average of the ratios of CD9, CD63, and CD8". Please describe how this is calculated exactly. The same for relative EV/ Albumin and relative EV/ApoB-100.

2) Please also calculate the relative EV yields, relative EV/ Albumin, and relative EV/ApoB-100 for the density gradient centrifugation experiment. Also, Figure 3 is a bit hard to follow due to the distance between the bars.

3) For Figure 5, it will be great to draw lines between 2 replicas to provide a sample-to-sample comparison.

4) Different fractions were selected for SEC (fractions 7-10), DMC (fractions 9-12), and TMC (fractions 9-12). Please provide the reason and the potential influence of the fairness of the comparison.

5) Figure 5—figure supplement 2, the decrease of EV yield cross SEC, DMC, and TMC is different for CD9, CD63, and CD8 (increasing in DMC). What is the reason?

6) The mechanism for the better performance of TMC needs to be discussed.

---

## [Author Response]

Essential revisions:1) Please observe that, due to the methodological nature of your study, it is extremely important to make sure that you consider the experimental suggestions provided by reviewers 1-3;2) We expect this manuscript to be of use to a wide audience of biologists since the EV community is growing rapidly. Please also consider the suggestions for improving the manuscript's presentation and discussion depth.

Thank you to the reviewers for their suggestions. We address the specific comments below.

Reviewer #1 (Recommendations for the authors):The paper can be very useful, but can be improved by addressing a few comments:1. 'We then decided to take advantage of the property that ApoB-100 is positively charged (12), while EVs are generally negatively charged (13) to separate EVs from lipoproteins.'LDL particles, as particles, are negatively charged. See, for example:• La Belle M, Blanche PJ, Krauss RM. Charge properties of low-density lipoprotein subclasses. J Lipid Res. 1997 Apr;38(4):690-700. PMID: 9144084.• Reynolds L, Mulik RS, Wen X, Dilip A, Corbin IR. Low-density lipoprotein-mediated delivery of docosahexaenoic acid selectively kills murine liver cancer cells. Nanomedicine (Lond). 2014 Jul;9(14):2123-41. doi: 10.2217/nnm.13.187. Epub 2014 Jan 7. PMID: 24397600; PMCID: PMC4156561.The paper cited shows the charge of certain ApoB100 peptides!LDL, VLDL, etc., contain many other components, not just ApoB100.

It is certainly true that ApoB-100 is not the only protein component of lipoproteins. We initially chose ApoB-100 as it was the main protein we detected by mass spectrometry after isolating EVs from plasma using SEC. This makes sense, as ApoB-100 is present in several lipoproteins that overlap in size with EVs (1). Regarding the question of whether ApoB-100 can be depleted based on cation exchange, it was not clear that this would work just based on the knowledge that there are positively charged peptides in ApoB-100, but we were inspired by previous work reporting the development of the dual mode chromatography columns (2). We were able to confirm the ability of cation exchange resin to deplete lipoproteins containing ApoB-100 with our ApoB-100 Simoa assay.

2. 'We compared EV isolation from plasma using SEC, DMC, and TMC columns. We first used electron microscopy to image EVs from each column and found that TMC led to EVs of the highest purity (Figure 4B).'The majority of the particles present on TEM images are lipoproteins, more likely VLDL or Chylomicrons. One can notice more dense round shaped particles, different from 'doughnut-shaped' EVs. For example, Figure 4B TMC column TEM image has a single EV, while eleven lipoprotein particles are visible.

We agree that the majority of particles in by TEM are lipoproteins and not EVs. However, the TEM demonstrates that there is an increase in purity of EVs relative to lipoproteins for TMC/DMC compared to SEC using a technique other than Simoa. Despite the significant reduction of ApoB-100 in TMC, there is still ApoB-100 present. The ApoB-100 can be further reduced by increasing the amount of cation-exchange resin in the TMC column. We provide additional data in Figure 4-supplemental figure 1 showing how varying the volumes of the Fractogel cation exchange resin and the Capto Core 700 resin changes the levels of tetraspanins, albumin, and ApoB-100. Increasing the amount of cation-exchange resin reduces ApoB-100, but also leads to a decrease in tetraspanins.

3. 'Using TMC, we were able to detect 780 proteins from EVs isolated from only 1 ml of plasma (Supplementary table 1). These results demonstrate the advantage of using TMC for deep proteomics analysis using a small sample volume.'According to the dataset provided in the supplementary materials, ApoB100 remains the most abundant protein, as well as other lipoproteins, including the ones specific for High-density lipoproteins. Also, a lot of free proteins were identified. In my experience, 780 proteins after albumin depletion are not as high for raw plasma proteomics analysis.

Both the electron microscopy (discussed above) and mass spectrometry analysis show that ApoB-100 is still present after TMC. We think it is likely impossible to fully separate EVs from lipoproteins and free proteins, but by increasing the ratio of EVs to lipoproteins and free proteins, as we have done with the TMC column, the depth of coverage for proteomics can be greatly increased. Our focus here was to provide a framework for quantitatively comparing EV isolation method yield and purity (with respect to both lipoproteins and free proteins); mass spectrometry is one application where having high purity EVs is important. The optimal EV isolation method will vary for different applications and will be highly dependent on the amount of starting material necessary for the technique being used. In mass spectrometry, different instruments have different requirements for the amount of input protein material necessary, so the optimal isolation technique will be instrument-dependent. We provide additional data in Figure 4-supplemental figure 1 (discussed above) showing how increasing the amount of cation-exchange resin allows for further depletion of ApoB-100. We did not, however, test all the different TMC variations using mass spectrometry. We are currently working on optimizing mass spectrometry for EVs from plasma, but this work is beyond the scope of this paper, as it also depends on many parameters downstream of EV isolation. Nonetheless, 780 proteins is high for a single-step technique from a starting volume of only 1 mL. For comparison, one study used a three step protocol (PEG precipitation followed by iohexol gradient fractionation and SEC) to enrich EVs from 1 mL plasma and reported detecting 250 proteins (3). Another group reported detecting 1187 proteins from plasma using ultracentrifugation, density gradient, and then SEC, but started with 40-80 mL plasma (4). We’ve added this information to the Discussion section.

4. 'In this work, we developed methods to separate EVs from both lipoproteins and free proteins in plasma based on our ability to measure proteins associated with these different components using ultrasensitive assays.'This statement is not confirmed by proteomics analysis and TEM images. Authors can use the more accurate term 'enriched EVs' compared to lipoproteins. Also, depletion of chylomicrons and HDL were not shown.

Thank you for the suggestion. We modified the wording of this sentence and replaced “separate” with “enrich.” We did not specifically analyze cholymicrons or HDL in this study. As mentioned in the Discussion section, Simoa assays for protein components of lipoproteins other than ApoB-100 (such as Apo-A1 for HDL) could be developed and an identical analysis framework could be employed. In this study, we decided to focus on the most abundant protein component of lipoproteins found in our mass spectrometry analysis, ApoB-100.

Reviewer #2 (Recommendations for the authors):I have a few suggestions for the authors' consideration:It may be more accurate to describe the free proteins and lipoproteins as impurities (i.e., present at the outset, but not intended to be present in the final preparation), rather than contaminants (i.e., introduced inadvertently)

We’ve changed “contaminants” to “impurities.”

The introduction could be more laser-focused on plasma since that's the focus of the work, and the lipoprotein issue may be greatly magnified compared to in e..g, urine

We’ve added this clarification to the last paragraph of the introduction.

A few more words about the principle of the Simoa assay could be useful to the reader

We’ve added two sentences about Simoa to the Results section.

"These results demonstrate the advantage of using TMC for deep proteomics analysis using a small sample volume." Some comparison to the prior state of the art would bolster this conclusion.

We have added these comparisons (as mentioned in response to one of the comments from Reviewer #1) to the Discussion section.

Beyond the relative yields, can synthetic EV spike-ins be used to calculate the absolute yields?

EV spike-ins from cell lines can be used, but we don’t think this is a way to provide absolute quantification since existing EV analysis techniques are unable to provide absolute quantification of these cell-culture derived EVs (as there are non-EV particles in cell culture media as well). Additionally, using EV spike-ins assumes that these EVs have very similar properties to plasma EVs, which may not necessarily be the case.

Figure legend for Figure 2: The EV yield is calculated "by averaging the ratios of CD9, CD63, and CD81." I think this be explained in clearer terms. This is the average of what is taken in ratio to what? This comment applies to a variety of other places in the paper where ratios are mentioned but without precise definitions of the numerator and denominator."Simoa measurements in the designated fractions for CD9, CD63, and CD81 are taken as a ratio relative to measurements of these proteins from diluted plasma and these three rations are then averaged to calculate recovery." Here again, I think this can be explained with a clearer description of what is in the numerator and denominator of these ratios (also a typo, "rations")It would be helpful if the authors wrote down the calculation they're referring to here: "by averaging the ratios of each of the tetraspanin levels between conditions" The numerator and denominator in the ratios are not as clear as they ideally could be.

We have added a paragraph about the EV yield and purity calculations to the Methods section and also expanded the explanation in the Results section.

A challenge is that after the removal of the main known impurities, whether additional proteins identified in proteomics experiments are in the vesicles or free proteins is not simple to distinguish. How does the relative quantitation of proteins after the application of the new column compare with plasma proteomics performed without enrichment for extracellular vesicles? Is the rank order of abundance similar?

We agree that distinguishing whether a given protein in an EV preparation is truly in EVs or stuck to the outside remains a major challenge in the field. One way to approach this challenge is with proteinase protection assays, but significant optimization is required to ensure both sufficient proteinase activity and full inactivation of proteinase before analyzing the protected EV contents. Although comparing the proteomes of total plasma and plasma EVs was beyond the scope of this study, such a comparison (using data from several published studies) has been performed before (5). In that comparison, fewer than half of the proteins detected between total plasma and plasma EVs were shared, and for those proteins that were shared, the correlation of protein abundances was relatively poor.

The authors have shared all the relevant data, as well as additional resources that will be useful to the scientific community.Reviewer #3 (Recommendations for the authors):1) The "relative EV yields" was used to quantify the amount of EVs across different methods. The authors described it as "the average of the ratios of CD9, CD63, and CD8". Please describe how this is calculated exactly. The same for relative EV/ Albumin and relative EV/ApoB-100.

We have expanded the explanation of how we calculate EV yield in the Results section and added a paragraph on this to the Methods section.

2) Please also calculate the relative EV yields, relative EV/ Albumin, and relative EV/ApoB-100 for the density gradient centrifugation experiment. Also, Figure 3 is a bit hard to follow due to the distance between the bars.

We have replotted Figure 3 to display each marker separately. We have also redone the density gradient centrifugation experiment to include SEC from the same sample of plasma, as well as a condition combining density gradient centrifugation and SEC. We have added Figure 3—figure supplement 1 to compare relative EV yields, EV/albumin and EV/ApoB-100 for these conditions.

3) For Figure 5, it will be great to draw lines between 2 replicas to provide a sample-to-sample comparison.

For the experiment in Figure 5, we started with 16 samples (1 mL each) of the same pooled plasma. We ran eight of these samples using the automated SEC device and the other eight samples performing SEC manually. Each plotted point for a given marker is an average of two Simoa measurements (technical replicates). We reworded description of this experiment in the Results section and the figure caption to clarify this.

4) Different fractions were selected for SEC (fractions 7-10), DMC (fractions 9-12), and TMC (fractions 9-12). Please provide the reason and the potential influence of the fairness of the comparison.

As the SEC column has 10 mL Sepharose CL^-^6B resin, and the DMC and TMC columns have 2 mL additional resin below the 10 mL Sepharose CL^-^6B resin, the difference in column volume leads to a change in where the EVs elute. We decided that taking four 0.5 mL fractions for each column would be best for directly comparing them. To decide which fractions to take, we analyzed the tetraspanins, albumin, and ApoB-100 in each individual fraction (as indicated in Figure 4-supplement 2).

5) Figure 5—figure supplement 2, the decrease of EV yield cross SEC, DMC, and TMC is different for CD9, CD63, and CD8 (increasing in DMC). What is the reason?

We find a difference in yield for the different tetraspanins between SEC and DMC/TMC. One possibility could be a difference in charge between different tetraspanins or subpopulations of EVs containing different tetraspanins.

6) The mechanism for the better performance of TMC needs to be discussed.

The TMC has improved performance in terms of free protein depletion relative to the SEC because the Capto Core 700 multimodal resin irreversibly binds free proteins that co-elute in the EV fractions. The TMC column has improved performance relative to SEC in terms of lipoprotein depletion for the same reason as the DMC column. The cation exchange resin binds and retains positively charged ApoB-100-containing lipoproteins. We have expanded the explanation of why TMC provides EVs of higher purity than SEC in the Discussion section.